# Gradual Learning for One-Shot Segmentation of Slice Stacks

**Johann Christopher Engster**[1]        JOHANN.CHRISTOPHER.ENGSTER@IMTE.FRAUNHOFER.DE
**Nele Blum**[1]        NELE.BLUM@IMTE.FRAUNHOFER.DE
**Thorsten M. Buzug**[1,2]        THORSTEN.BUZUG@IMTE.FRAUNHOFER.DE
**Maik Stille**[1]        MAIK.STILLE@IMTE.FRAUNHOFER.DE

[1] *Fraunhofer IMTE, Fraunhofer Research Institution for Individualized and Cell-Based Medical Engineering, Lübeck, Germany*
[2] *University of Lübeck, Institute of Medical Engineering, Lübeck, Germany*

**Editors:** Under Review for MIDL 2024

## Abstract

Diagnostic imaging modalities like magnetic resonance imaging (MRI) or computed tomography (CT) are crucial for medical and industrial inspection. However, labeled datasets are not always available for segmentation of rare cancer types or other defects. Therefore, a new training strategy named gradual learning is proposed for one-shot segmentation, thus requiring only one labeled example slice. A segmentation network trained on this input generates suitable pseudo labels in a local neighborhood, with the quality degrading with distance. These adjacent pseudo labels can be incorporated into the training process repeatedly, to process the unlabeled slices step-by-step. Experiments were conducted on MRI head scans for skull-stripping. A total of 30 models were trained using gradual learning, receiving one scan with one annotated slice each. On a separate test set ($n = 30$ scans), the mean intersection over union (mIoU), averaged over all models, increased from 0.885 to 0.935 using gradual learning compared to training without it. When trained with the ground truth (GT) of the same slices instead the models achieved a 0.955 mIoU.

**Keywords:** Segmentation, Semi-Supervised Learning, One-Shot Learning

## 1. Introduction

In medical imaging, artificial intelligence has achieved considerable success in recent years. One of the central application fields is segmentation, which, for instance, helps physicians with the diagnosis (Liu et al., 2021). Despite the notable achievements of learning-based methods in this area, many approaches are limited to supervised learning, which requires a large amount of labeled training data. The acquisition of this data is very time-consuming and not always achievable. Our developed method, however, addresses this issue and can be applied to any slice-based 3D image data. By developing a semi-supervised method, only a single labeled slice from the entire 3D slice stack is required to produce a full segmentation. This is achieved using a one-shot segmentation algorithm, as exemplified on MRI head scans to perform skull-stripping (Hoopes et al., 2022) in the following.

## 2. Methods

For most learning processes, it is helpful to gradually increase the task difficulty. Even if it were possible to go from zero to hero, it might be more advantageous to take a step-wise (Konishi et al., 2021), or in this case slice-wise, learning approach. The fundamental concept is shown in Algorithm 1. Given a stack of images, a segmentation network can be trained on one manually annotated starting slice. The trained network is then applied to

the remaining slices which generates pseudo labels (Ito et al., 2019). However, if the input data is less known to the model, the quality of the pseudo labels may deteriorate. Compared to (Grewal et al., 2023), it is not feasible to use entropy for uncertainty estimation because an ensemble trained on a single example might exhibit correlated uncertainties. Therefore, for this slice-wise approach, we assume the following: The greater the real-world distance between the training slice and the input slices, the higher the uncertainty. Thus, by focusing only on the local neighborhood of the training slice, we leverage their high similarity. After each training run, the network generates pseudo labels for the adjacent slices, which are then added to the training set.

---

**Algorithm 1:** Gradual Slice-Wise Training

---

// MRI slices, labeled start-slice at index i, and hyperparameters;
**Input:** slices, start_label, start_i, num_runs, step_size
**Output:** data
data[start_i] ← [start_label, slices[start_i]];
**for** $run \leftarrow 1$ **to** $num\_runs$ **do**
    trainer ← Trainer();
    trainer.train(data);
    **for** $i \leftarrow 1$ **to** $run \times step\_size$ **do**
        data[start_i ± i] ← [trainer.predict(slices[start_i ± i]), slices[start_i ± i]];
    **end**
**end**

---

For proof of concept, the *Calgary-Campinas Public Dataset* (Souza et al., 2017) for skull-stripping, namely the segmentation of brain matter, was used. In total, 60 out of 359 scans were selected, 30 each for training and testing, each containing 100 to 150 slices with an input dimension of $256 \times 256$. The U-Net++ (Zhou et al., 2018) architecture was used to evaluate the gradual learning. For each training scan, a separate model was trained, using only a single annotated starting slice. Each model was then trained for ten runs, each with a step size of five and 10000 iterations, as outlined in Algorithm 1. Subsequently, each model was evaluated on all test scans.

## 3. Results and Discussion

Figure 1 shows different quantitative results for the test data set. The average achieved mIoU at each gradual run, compared to the performance of a second model that was trained using the GT labels instead of the pseudo labels, is depicted (1(a)). Figure 1(b) illustrates the average, slice-wise mIoU obtained for different gradual run numbers with pseudo labels.

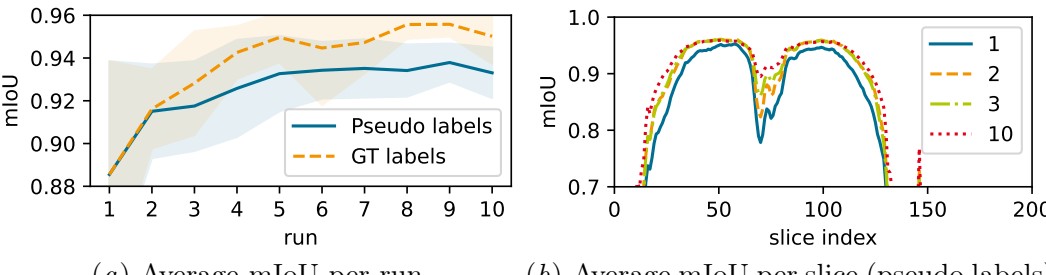

(a) Average mIoU per run.      (b) Average mIoU per slice (pseudo labels).

Figure 1: The average mIoU performance on the test set (30 scans).

The mIoU increased using both pseudo labels and GT for these slices. The gradually trained approach achieved a performance gain of about half that observed with the GT (an increase of +0.05 vs. +0.07) relative to the baseline (mIoU at run number 1). The first run had a high standard deviation, which emphasized a large number of uncertain slices. This uncertainty decreased over subsequent runs. Figure 1(b) clearly shows the increased performance per slice, as many slices, particularly around the middle (index $\approx 70$), were not segmented accurately after the first run. These slices, which typically depict the transition between brain hemispheres, contain challenging information. Gradual learning, however, efficiently bridged this gap.

Figure 2 shows pairs of input images and masks at run numbers 1, 5, and 10. The initial run was based solely on a single available starting slice. Run numbers 5 and 10 were additionally trained on the obtained pseudo labels or GT labels, respectively. The figure depicts both the distance from the starting slice and the achieved IoU for each slice.

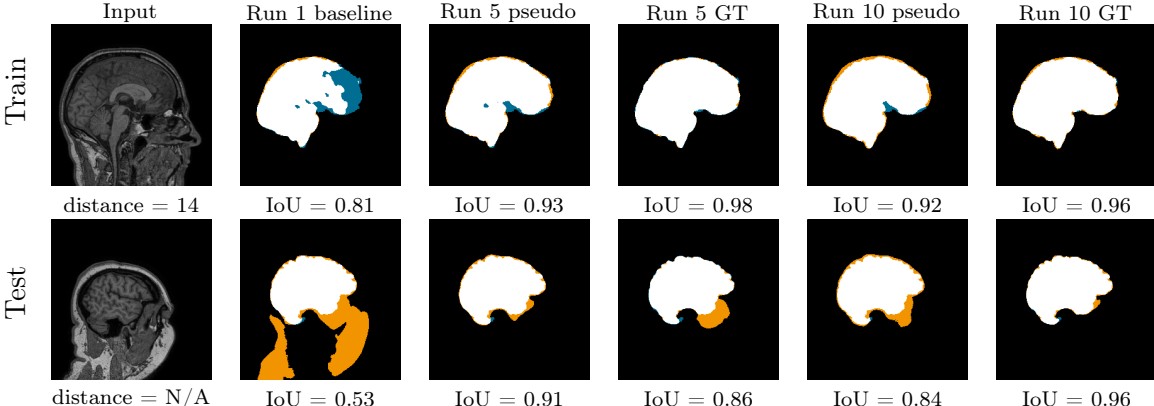

Figure 2: Different qualitative examples showing input images and the achieved masks after different runs for both the pseudo-label-trained and GT-trained networks. True positives $= \square$, false positives $= \blacksquare$, and false negatives $= \blacksquare$.

In Figure 2, the first run shows examples of false positives and false negatives, which were improved in subsequent runs. The best results from using pseudo labels for gradual learning occurred in run 5, which suggests a potential cumulative error that could degrade performance after too many runs.

## 4. Conclusion

This paper proposes a strategy for one-shot segmentation of slice-based imaging modalities. With this gradual learning, slices can be processed sequentially, leveraging the potential of the already acquired pseudo labels. Both the quantitative and qualitative evaluations are promising when compared to the baseline network that was trained without this gradual strategy. Notably, the proposed method has also achieved similar performance with other industrial and medical datasets. This method can be applied to the segmentation of 3D objects, provided there is a suitable initial annotation and the subsequent slices have high similarity. Future work may include incorporating active learning.

## Acknowledgments

Funded by the German Federal Ministry of Education and Research, FKZ 03XP0581B

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
