# OpenReview forum: "Gradual Learning for One-Shot Segmentation of Slice Stacks"
_MIDL.io/2024/Short_Papers — MIDL 2024 Short Papers_

### Official Review · Reviewer_B9Sn · 2024-04-24

**Confidence:** 5
**Final Rating:** 5

**Review:**

*** 23 Gradual Learning for One-Shot Segmentation of Slice Stacks
This submission proposes a novel method for one-shot segmentation of medical images, specifically MRI scans. The proposed method addresses the challenge of limited annotated data by leveraging recent advances in self-supervised learning, where the model learns from unlabeled data by creating its own supervision. While the submission demonstrates the effectiveness of gradual learning for MRI slice segmentation, it would be beneficial to strengthen the motivation for this approach in a clinical setting. Additionally, the dependency on slice orientation needs to be addressed, as variations in orientation can impact segmentation accuracy. For these reasons, the recommendation is towards Acceptance.

---

### Decision · Program_Chairs · 2024-04-26

Accept